# Augmenting the technology acceptance model with trust model for the initial adoption of a blockchain-based system



Ajay K. Shrestha[1], Julita Vassileva[1], Sandhya Joshi[2] and Jennifer Just[1]

[1] Department of Computer Science, University of Saskatchewan, Saskatoon, Saskatchewan, Canada
[2] Unaffiliated, Saskatoon, Saskatchewan, Canada

## ABSTRACT

**Background:** In the collaborative business environment, blockchain coupled with smart contract removes the reliance on a central system and offers data integrity which is crucial when the transacting parties rely on the shared data. The acceptance of such blockchain-based systems is necessary for the continued use of the services. Despite many extensive studies evaluating the performance of blockchain-based systems, few have focused on users' acceptance of real-life applications.

**Objective:** The main objective of this research is to evaluate the user acceptance of a real-life blockchain-based system (BBS) by observing various latent variables affecting the development of users' attitudes and intention to use the system. It also aims to uncover the dimensions and role of trust, security and privacy alongside the primary Technology Acceptance Model (TAM)-based predictors and their causal relationship with the users' behavior to adopt such BBS.

**Methods:** We tested the augmented TAM with Trust Model on a BBS that comprises two subsystems: a Shopping Cart System (SCS), a system oriented towards end-users and a Data Sharing System (DSS), a system oriented towards system administrators. We set research questions and hypotheses, and conducted online surveys by requesting each participant to respond to the questionnaire after using the respective system. The main study comprises two separate sub-studies: the first study was performed on SCS and the second on DSS. Furthermore, each study data comprises initial pre-test and post-test data scores. We analyzed the research model with partial least square structural equation modelling.

**Results:** The empirical study validates our research model and supports most of the research hypotheses. Based on our findings, we deduce that TAM-based predictors and trust constructs cannot be applied uniformly to BBS. Depending on the specifics of the BBS, the relationships between perceived trust antecedents and attitudes towards the system might change. For SCS, trust is the strongest determinant of attitudes towards system, while DSS has perceived privacy as the strongest determinant of attitudes towards system. Quality of system shows the strongest total effect on intention to use SCS, while perceived usefulness has the strongest total effect on intention to use DSS. Trust has a positive significant effect on users' attitudes towards both BSS, while security does not have any significant effect on users' attitudes toward BBS. In SCS, privacy positively affects trust, but security has no significant effect on trust, whereas, in DSS, both privacy and security have significant effects on trust. In both BBS, trust has a moderating effect on privacy that correlates with attitudes towards BBS, whereas security does not have any mediating role

Corresponding author
Ajay K. Shrestha,
ajay.shrestha@usask.ca

between privacy and attitudes towards BBS. Hence, we recommend that while developing BBS, particular attention should be paid to increasing user trust and perceived privacy.

## INTRODUCTION

Blockchain technology has influenced significantly the financial world with its first application in the form of cryptocurrencies such as bitcoin (*Nakamoto, 2008*). After a decade long development phase, it has now exhibited high potential for a broader diffusion across many other industries such as healthcare, agriculture, tourism and research fields (*Bullock & Bannigan, 2016*; *Feng, 2016*; *McGhin et al., 2019*; *Shrestha & Vassileva, 2016*, *2018b*). Blockchain technology is likely to disrupt many of the traditional centralized business models because of its being decentralized, immutable, tamper-proof and transparent processes (*Shrestha & Vassileva, 2018a*; *Swan, 2015*). Many blockchain systems also support smart contracts that encode the business logic into an autonomous self-executing piece of a program and are also deployed on the blockchain. A Smart contract stores the rules which negotiate the terms of the contract, automatically verifies the contract and executes the agreed terms whenever it is triggered by the target collaborator or by the responsible process from another smart contract. Blockchain coupled with smart contract technology removes the reliance on the central system between the collaborators and the transacting parties (*Shrestha, Deters & Vassileva, 2017*). However, blockchain is not a silver bullet that can be incorporated into any business use case. It is particularly important to identify and conduct careful analysis and evaluation of different factors affecting the collaborative business model that is built on the top of blockchain and smart contracts technologies. Furthermore (*Prashanth Joshi, Han & Wang, 2018*) argued in their comprehensive survey that numerous privacy and security-related issues have risen while adopting blockchain-based applications (*Kshetri, 2017*), based on their findings, suggested that although blockchain supports peer-to-peer security, the decentralized application itself is vulnerable to security breaches and privacy infringements.

As suggested by *Cunningham (1967)*, the evaluation process is crucial in studying the user perception of the adoption of new information technology services. The Technology Acceptance Model (TAM) proposed by *Davis (1989)* has been used widely in the literature to examine whether users understand the underlying technology and can competently use the services (*Granić & Marangunić, 2019*). In many studies, researchers extend TAM by adding external constructs depending upon the contexts to explain the critical relationship between customers and their adoption of the new technology (*Melas et al., 2011*). With the rapid development of the use cases of blockchain in recent years, a few studies have already been conducted considering the user acceptance of an abstract

blockchain-based system (*Folkinshteyn & Lennon, 2016*; *Kern, 2018*; *Shin, 2019*; *Shrestha & Vassileva, 2019a*). Although numerous extensive systematic studies have been conducted on evaluating the performance of blockchain-based systems (*Shrestha, Vassileva & Deters, 2020*), to the best of our knowledge, no study has been conducted in the context of users' acceptance of real-life blockchain-based applications except for bitcoin as financial technology (*Folkinshteyn & Lennon, 2016*). Previous works have evaluated user acceptance of the blockchain-based **prototype system** using an extended Technology Acceptance Model (TAM) in *Kern (2018)*, *Shrestha & Vassileva (2019a)* and the trust model in *Shin (2019)*. The previous studies suggest that the blockchain-based system will be accepted if it is perceived as trustworthy, convenient and useful (*Kern, 2018*; *Shin, 2019*).

The major contribution of this study is that it expands the previous work by conducting a new user study on a real-life blockchain-based system (BBS), described in *Shrestha, Joshi & Vassileva (2020)*. This study presents the augmented TAM by incorporating additional constructs—Trust, Perceived Security and Perceived Privacy—in technology adoption study and presents the total effect and mediation analyses. The findings are informative and potentially useful for designing new blockchain-based systems.

The BBS of our study is the general-purpose blockchain-based system that provides a solution to four important problems: private payment, ensuring privacy and user control, and incentives for sharing. This BBS was constructed for the online shopping cart which also allows customers to connect to the seller directly and share personal data without losing control and ownership of it. This BBS has two subsystems- a customer-specific shopping cart system (SCS), and a company-specific data sharing system (DSS). SCS allows customers to set their data sharing preferences and deploy them via smart contracts, which gives customers full transparency over who accesses their data, when and for what purpose, specifies the purposes of data sharing, which kinds of data can be shared, which applications or companies can access their data and provide an incentive to them for sharing their data in terms of micropayment as stated in the contract. Similarly, DSS allows companies to check data integrity, get tamper-proof records and proof of existence of every transaction while sharing data in the consortium blockchain network. Therefore, the BBS used in the study is a very novel decentralized application that covers the aspects of both the customer and company. So, its in-depth analysis to examine all those factors of the Trust model and the TAM indicators that mostly affect the user acceptance of the BBS is crucial to provide an opportunity for a broad debate and perspective on potential uses of blockchain and smart contract technologies for the eCommerce domain along with other different important industries such as healthcare, agriculture, tourism and research fields.

Therefore, our current study is based on the user evaluation of the blockchain-based SCS and DSS, before and after using those sub-systems by the selected participants, using the validated constructs of the TAM and the Trust model. This new augmented model incorporates both classical TAM with perceived ease of use, perceived usefulness and quality of system, and Trust model with security and privacy variables, and it can be applied to evaluate the acceptance of the general blockchain- and smart contracts-based

systems. The present study, using the partial least square structural equation modeling on augmented TAM, hypothesizes and validates various causal relationships to observe the statistical significance between the constructs of interest and intention to use the BBS.

The remainder of the paper is organized as follows: "Background" provides some background of the BBS and augmented TAM model. "Research Model and Hypotheses" presents the research models and hypotheses. The research methodology with measurement and structural models is presented in "Materials & Methods". "Results" focuses on the brief analysis of the results and "Discussion" provides the discussion. The limitation of the study is presented in "Limitations". Finally, "Conclusions" concludes the paper.

## Background

This section provides background information about the Blockchain-Based System used in the study described in this paper, the Technology Acceptance Model, and the models of Privacy, Security and Trust used to predict software systems' adoption by users.

### Blockchain-Based System (BBS)

The term BBS for a general blockchain-based system was initially used in *Jun (2018)* without any detailed explanation. BBS in our study represents the blockchain-based service that we have developed with an engineering-oriented approach to address trust-aware business processes in an e-commerce domain, in the context of an online shopping cart system (*Shrestha, Joshi & Vassileva, 2020*). The requirements for the BBS are:

- To enable companies to increase trust in their products and supply chains.
- To offer direct payment with native Ethereum tokens thereby enabling privacy and confidentiality.
- To create proof of the existence of every transaction.
- To give the users full transparency over who accesses their data, when and for what purpose.
- To enable companies to share customers' data among others in the consortium network.
- To provide incentives to customers in real-time for sharing their data.

This BBS has a 3-tier architecture (*Fernandez et al., 2008*) employing *Spring Boot* (https://spring.io/projects/spring-boot) and *React* (https://reactjs.org/) as the main building technologies. The system uses permissioned MultiChain as a solution to both on-chain and off-chain data storage, encryption, hashing and tracking of data, together with Ethereum. Ethereum is used for access control and enabling transactions with *ethers* that allow users to shop online with all the transactions stored in the blockchain and get incentives for permitting to share their data as they specify in the smart contracts. Figure 1 presents the interaction among the customer (data provider) and other e-commerce companies/apps (data consumers) of the BBS. The system comprises two subsystems: Shopping Cart System (SCS) and Data Sharing System (DSS). SCS is used in the online shopping cart enterprise. It has a payment mechanism supporting cryptocurrency, *ether*

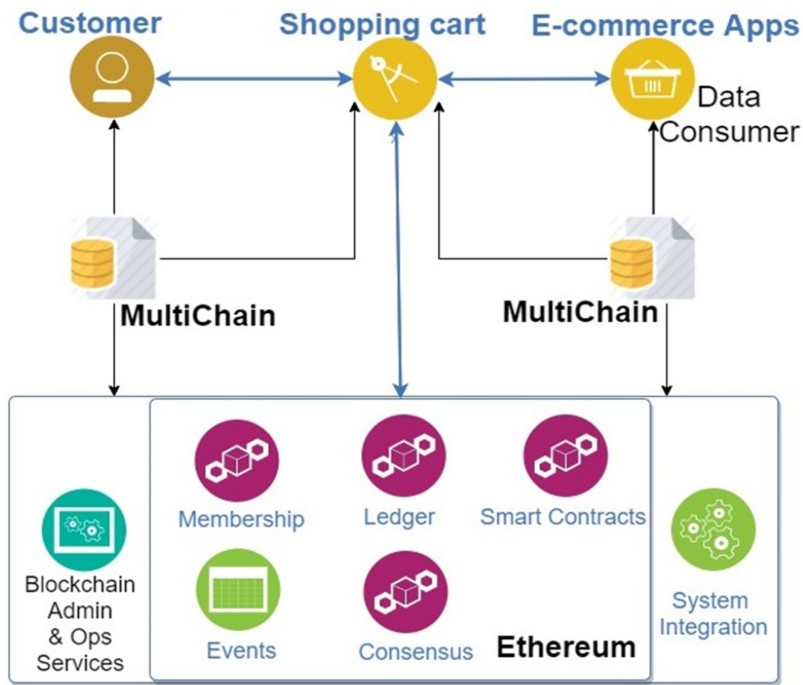

**Figure 1 Blockchain-Based System of the current study (BBS).** © 2020 IEEE.

and manages the mutual agreement between customers and enterprise through smart contracts. SCS automatically registers the immutable timestamped information about the transactions that acts as proof of existence and can be useful to settle any disputes between the stakeholders in the future. Moreover, SCS deploys smart contracts that allow customers to provide their data sharing preferences on a template form without needing them to write the code for the smart contracts. The smart contracts support users in the following ways (*Shrestha & Vassileva, 2019b*):

- Give users full transparency over who accesses their data, when and for what purpose.
- Allow users to specify the purposes of data sharing, which kinds of data can be shared, and which applications or companies can access the data.
- Provide an incentive to users for sharing their data (in terms of payment for the use of the data by applications, as specified by the contracts).

DSS is used for sharing user data among the companies, that provide the shopping cart system to the customers. DSS allows enterprises to form a consortium blockchain network in the MultiChain environment so that user data are only shared with the particular node, that has been given the data access permission, as defined in the smart contracts when deployed by customers on SCS. DSS offers tamper-proof encrypted data storage, publication and provenance mechanisms with a transparency of the event log mechanism in collaborative processes where different enterprises use published/shared data.

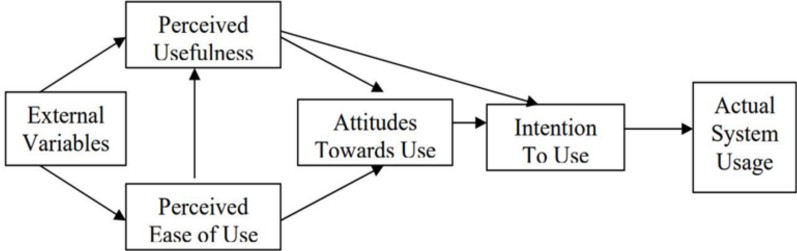

**Figure 2 Classical Technology Acceptance Model (TAM).**

## Augmented Technology Acceptance Model

The classical Technology Acceptance Model (TAM) as shown in Fig. 2 was based on the Theory of Reasoned Action (*Fishbein & Ajzen, 1975*) in social psychology, which claims that behavioral intention is a strong indicator of actual behavior. The TAM has been used as a conceptual framework in many studies of the potential users' behavioral intention to use a particular technology. The behavioral intention is defined as "the degree to which a person has formulated conscious plans to perform or not perform some specified future behavior" (*Warshaw & Davis, 1985*). The classical TAM focuses on using technology, where perceived ease of use (PEOU) and perceived usefulness (PU) are two design attributes or antecedent to influence user acceptance behavior. PEOU is defined as the degree to which a person believes that using a particular system would be free of effort. PU is the degree to which a person believes that using a particular system would enhance his or her job performance. TAM hypothesizes that the actual use of the system is determined by behavioral intention to use (ITU), which is the degree to which a person has behavioral intention to adopt the technology. ITU is in turn influenced by the user's attitude towards use, perceived usefulness and perceived ease of use of the system. Attitude towards use is the degree of belief to which a person uses the system as guided by valuations (*Shin, 2019*; *Shin, 2017*).

TAM is widely used to understand how users come to accept and use information technology. However, there is no existing literature on using TAM in the context of real-life blockchains and smart contracts-based applications, indicating a significant gap in the knowledge. To fill this research gap in the existing literature, this study applies the augmented TAM with trust model to the BBS that we implemented, with participants who actually used the system before answering the survey questionnaires. Our study also uncovers the individual mediating effects of trust, security and perceived usefulness.

In classical TAM, the main design constructs such as perceived ease of use and perceived usefulness have shown significant influence on the behavioral intention of the user to adopt the information systems (*Davis, 1989*), and the latest study by *Shin (2019)* shows the necessity of considering the Trust-Security-Privacy factors in the decision model of the blockchain-based-solution adoption. So, we adopted the partial least square structural equation modeling (PLS-SEM) analyses on the augmented TAM as it is a useful technique to estimate complex cause-effect relationship models with latent variables and

we aimed to model the latent constructs under conditions of non-normality and small sample sizes (*Wong, 2013*).

Many researchers often extend TAM by adding external constructs because classical TAM often does not capture many key factors specific to the context of the technology (*Melas et al., 2011*). Quality of system (QOS) (*Koh, Prybutok & Ryan, 2010*), trust (T) (*Wu & Chen, 2005*), behavioral control (*Bhattacherjee, 2000*) are some of the constructs that have been added as influential variables to user acceptance of the information technology and are therefore inevitable for evaluating a novel system, BBS as in this current study. Although in the software engineering domain, security and privacy are regarded as part of QOS, in this study, we have presented perceived security and perceived privacy as separate constructs. *DeLone & McLean (1992)* refers to QOS as the technical details of the system interface and system's quality that produces output response such that the technology attributes singularly or jointly influence user satisfaction. Hence, it is assumed that the QOS affects user satisfaction and that directly or indirectly through PU, affects users' intention to use the system (*DeLone & McLean, 1992*; *Shrestha & Vassileva, 2019a*).

Moreover, perceived privacy and perceived security have critical roles in the acceptance of the technologies as the prior research suggest they have a significant effect on users' attitudes that positively influence their intention to use the technologies (*Amin & Ramayah, 2010*; *Roca, García & de la Vega, 2009*; *Shin, 2010*).

## Multidimensionality of privacy

Privacy is defined as the right to be let alone (*Warren & Brandeis, 1890*). Furthermore, privacy has been considered as the right to prevent the disclosure of personal information to others (*Westin, 1968*). Later, privacy has been known to be not just unidimensional (*Burgoon et al., 1989*; *DeCew, 1997*) as it includes informational privacy along with accessibility privacy, physical privacy and expressive privacy.

- Informational privacy – "how, when, and to what extent information about the self will be released to another person" (*Burgoon et al., 1989*; *DeCew, 1997*), e.g., the user is asked for too much personal information while using online services.
- Accessibility privacy- "acquisition or attempted acquisition of information that involves gaining access to an individual" (*DeCew, 1997*), e.g., the user's contact (address, phone or email) information might be left in the old system.
- Physical privacy- "the degree to which a person is physically accessible to others" (*Burgoon et al., 1989*) e.g., viewing user screen in an unauthorized way.
- Expressive privacy- "protects a realm for expressing one's self-identity or personhood through speech or activity" (*DeCew, 1997*). It restricts extrinsic social control over choices and improves intrinsic control over self-expression, e.g., user data may be inappropriately forwarded to others.

*Introna & Pouloudi (1999)* developed a framework of principles for the first time to study privacy concerns while exploring the interrelations of interests and values for various

stakeholders. The study has identified that different users have distinct levels of concern about their privacy. *Smith, Milberg & Burke (1996)* developed a scale for the concern for privacy that measured unidimensional aspects of privacy such as collection, errors, secondary use, and unauthorized access to information factors. *Malhotra, Kim & Agarwal (2004)* also presented a model to consider multiple aspects of privacy such as identifying attitudes towards the collection of personally identifiable information, control over personal information and awareness of privacy practices of companies gathering personal information. However, all these studies just focused on the informational privacy, so the scales to measure privacy were also based on a unidimensional approach and were not even validated. Furthermore, the issue regarding the benefit to giving up privacy such as offering personalization, enhanced security etc. was not addressed by those studies.

Hence, to address the multidimensionality of privacy, it is particularly important to consider privacy-related behaviors while studying privacy concerns and user attitudes towards privacy in BBS. The constructs presented in a study by *Buchanan et al. (2007)* are validated and considered both privacy concerns and user behavior models. The behavioral items include general caution and technical protection of privacy. Attitudinal item includes privacy concern. The authors found that privacy concern correlates significantly with a general caution, but not significantly with the technical protection factor. Furthermore, perceived privacy, which is the attitudinal privacy or privacy concern undoubtedly plays a critical role in user accepting technologies (*Hoffman, Novak & Peralta, 1999*; *Poon, 2008*). It sheds light on the possibility of unauthorized use and access to the personal and financial information of the users by the companies that they are intending to use the service of *Dwyer, Hiltz & Passerini (2007)*.

## Perceived security

Perceived security is the degree to which a user believes that the online service has no predisposition to risk (*Yenisey, Ozok & Salvendy, 2005*). The protected financial and personal information may get compromised by theft and fraudulent activities leading to vulnerability on the internet. Because of this, a sense of security becomes a major concern for the customers to handout their details on the network (*Gefen, 2000*; *Shrestha, 2014*; *Wang, Lee & Wang, 1998*). Perceived security here does not only mean technical security but the user's subjective feeling of being secured in the network (*Roca, García & de la Vega, 2009*). Authors (*Linck, Pousttchi & Wiedemann, 2006*) have argued that a lack of subjective security in the user's mind will create hesitation to use systems.

## Trust as mediating factor

Trust is an important contributing factor for users to do a certain task that can make them vulnerable and yet hope the service provider on the other end to fully comply with the set of protocols to complete a transaction (*Dwyer, Hiltz & Passerini, 2007*) and eventually develop a new relationship (*Coppola, Hiltz & Rotter, 2004*; *Jarvenpaa & Leidner, 1999*; *Piccoli & Ives, 2003*). In a virtual environment, as the users do not have any control over the outcome of their actions, trust becomes one of the prime factors for them to ground some firm belief in the reliability to engage with the other party (*Hoffman, Novak &*

*Peralta, 1999*). In e-commerce, when information is disclosed, users tend to trust more the service provider (*Metzger, 2004*) resulting in users being free of doubts and are more likely to engage with the other party (*Hoffman, Novak & Peralta, 1999*). Research has shown that trust has a positive significant impact on attitude and intentions to use systems (*Papadopoulou, 2007*). With greater trust, users question less the authenticity of online services.

The user acceptance behavioral model, as presented by *Rios, Fernandez-Gago & Lopez (2017)*, *Shin (2010)* for theoretical social network services, is also useful for conceptualizing the role of perceived security, perceived privacy (privacy concern from attitudinal privacy) on user trust. Their findings revealed that perceived security has a moderating effect on perceived privacy that correlates significantly with trust the user can have on the system.

## Related work

Numerous studies have been conducted to examine the factors that determine the acceptance of information technology in the context of an extended TAM and Trust model. We cover a cross-section of those studies that are related to our work.

To the best of our knowledge (*Folkinshteyn & Lennon, 2016*), conducted a very first user study with TAM in the context of the adoption of bitcoin as financial technology. Their findings revealed both positive and negative factors associated with the acceptance of bitcoin, the first real-life application of blockchain technology. They have also argued that the cryptocurrency offers borderless and efficient transactions with significant positive factors in PEOU and PU, giving users full control over their currency, however it is also extremely volatile with not being lenient of security breaches or errors (*Folkinshteyn & Lennon, 2016*). So, it has both risks and benefits that affect the overall adoption of the cryptocurrency. Their findings also suggested exploring other aspects beyond TAM variables to consider the underlying risk and trustworthiness constructs associated with the blockchain-based applications. Previous research by *Kern (2018)*, *Shin (2019)* on an abstract blockchain-based application model suggested that the blockchain-based system can be accepted if it has enough trust to sustain and is perceived as convenient and useful in the highly competitive market. Almost all of the existing research so far is limited to the blockchain-based prototype system using an extended TAM (*Kern, 2018*; *Shrestha & Vassileva, 2019a*) and Trust model (*Shin, 2019*). Our current study extends the research contribution of the prior study (*Shrestha & Vassileva, 2019a*) by conducting a new user study on the real-life blockchain-based system, BBS (*Shrestha, Joshi & Vassileva, 2020*).

Gefen et al. have previously explored a mixed model with TAM and Trust model to study the adoption of the on-line shopping setting (*Gefen, Karahanna & Straub, 2003*). Their model presented the use of the on-line system into both system attributes such as perceived usefulness and perceived ease of use and trust in e-vendors. Their model resulted in the integrative indication of the TAM and Trust constructs as good predictors for the output response, which was the behavioral intention to use the online shopping system.

Therefore, the current study adopts a similar model and presents it as augmented TAM which comprises an extended TAM and Trust model.

As online activities such as online shopping generate a plethora of real-time transactions of all kinds of assets and information, they are prone to security and privacy-related risks (*Roca, García & de la Vega, 2009*). A privacy issue mostly occurs with unwarranted access to the users' personal data, but that does not necessarily involve security breaches, which can happen with poor access control mechanisms in the system allowing malicious actors to control the system. However, both breaches are critical issues and they often exist together on the online services where users typically feel hesitant to provide private information over the internet (*Hoffman, Novak & Peralta, 1999*). (*Shin, 2010*) previously explored the statistical significance of security and privacy in the acceptance of social networking sites. Later, *Shin (2019)* presented the role and dimension of digital trust in the emerging blockchain context, where (*Siegel & Sarma, 2019*) has argued that it has not been investigated how privacy/security factors affect user's behavioral cognitive process of accepting the blockchain-based systems. This study, in addition to previous TAM validated constructs, explores the users' perception towards the security and privacy aspect of the BBS and their influence on intention to use the BBS by using the moderating effects of trust on attitudes towards system. Besides, the current research aims to answer the following research questions when exploring the relationship between different indicators of the augmented TAM with the trust model:

- RQ1: Which of the design attributes is/are the strongest antecedents of the attitudes towards BBS?
- RQ2: Which of the design attributes is/are the strongest antecedents of the intention to use BBS?
- RQ3: Is the influence of privacy on attitudes towards BBS mediated by both security and/or trust?
- RQ4: Is the influence of security on attitudes towards BBS mediated by trust?
- RQ5: Is the influence of ease of use/quality of system on intention to use BBS mediated by perceived usefulness?

## RESEARCH MODEL AND HYPOTHESES

Figure 3 presents the structural model with the main constructs and their associated structural paths. Fourteen research hypotheses are thus constructed for our research model based on the findings of the literature review presented in the previous section.

**Perceived Usefulness and Perceived Ease of Use** (*Davis, 1989*; *Davis, Bagozzi & Warshaw, 1992*)

H1: Perceived ease of use significantly influences the perceived usefulness of BBS.

H2: Perceived ease of use significantly influences the intention to use BBS.

H3: Perceived usefulness significantly influences the intention to use BBS.

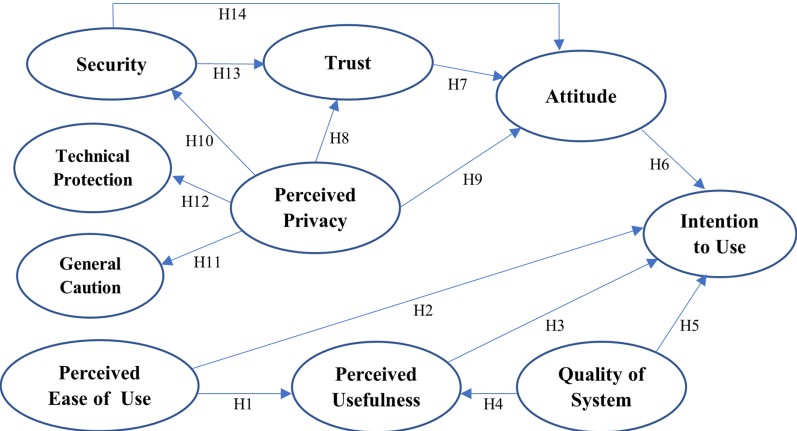

**Figure 3** An Augmented TAM with Trust Model.     

**Quality of System** (*Koh, Prybutok & Ryan, 2010*)
H4: Quality of system significantly influences the perceived usefulness of BBS.
H5: Quality of system significantly influences the intention to use BBS.
**Attitude Towards BBS** (*Shin, 2017*)
H6. Attitude towards BBS significantly influences the intention to use BBS.
**Trust** (*Dennis et al., 2012*); (*Jian, Bisantz & Drury, 2000*)
H7. Trust positively affects users' attitudes toward BBS.
**Perceived Privacy** (*Buchanan et al., 2007*)
H8. Perceived privacy has a positive effect on the users' trust in BBS.
H9. Perceived privacy has a positive effect on the users' attitudes toward BBS.
H10. Perceived privacy positively or negatively affects users' perceived security.
H11: Privacy concern positively affects users' behavior on general caution.
H12: Privacy concern positively affects users' behavior on technical protection.
**Perceived Security** (*Shin, 2010*)
H13. Perceived security positively affects users' trust in BBS.
H14. Perceived security positively affects users' attitude toward BBS.

The main study comprises two separate sub-studies: the first study was performed on the SCS and the second on the DSS. Furthermore, each study data comprises pre-test and post-test data scores. The pre-test defines the data collected from participants before they use the system, whereas post-test data is collected after participants use the system.

The pretest study can be considered as the study associated with the prototype model. Since, the present study follows the previous research work from *Shrestha & Vassileva (2019a)*, the pretests for the current study do not include the constructs from classical TAM as they were already evaluated in the previous study. So, the pretests of the current study do not present data for hypotheses H1–H6. The post-tests for both SCS and DSS do not have *behavioral privacy-general caution* and *-technical protection* constructs as they are only evaluated once, during the pre-test. So, the post-test data do not test hypotheses H11–H12.

## MATERIALS & METHODS

The present study was approved with delegated review by the University of Saskatchewan Behavioural Research Ethics Board (Beh-REB). The approval with reference number Beh # ID2106 was given for behavioural application/amendment form, consent form and survey questionnaire. We first conducted a pilot study with 14 participants from the Multi-User Adaptive Distributed Mobile and Ubiquitous Computing (MADMUC) Lab and quantitative research experts at the University of Saskatchewan to evaluate the feasibility, duration and improve upon the study design of our research approach. The participants in the pilot study provided feedback with their opinion of the survey in general. Based on the pilot test outcomes and the review of quantitative research experts, the final survey questionnaires were modified and restructured, and then the research model was empirically tested by collecting survey data. The design of the research instrument, sample organizations and sample demographics are described below.

### Research instrument design

We conducted online surveys through SurveyMonkey by requesting each participant to respond to the questionnaire on different constructs. The survey instrument is based on constructs validated in prior studies by *Buchanan et al. (2007)*, *Davis (1989)*, *Davis, Bagozzi & Warshaw (1992)*, *Dennis et al. (2012)*, *Jian, Bisantz & Drury (2000)*, *Koh, Prybutok & Ryan (2010)*, *Shin (2010*, *2017)* and adapted in the context of our research model. The instrument consists of 6 items for perceived ease of use, 6 items for perceived usefulness, 4 items for quality of system, 3 items for perceived enjoyment, 4 items for intention to use, 3 items for perceived security, 9 items for trust, 4 items for attitudinal privacy (perceived privacy), 4 items for behavioral privacy-general caution, 4 items for behavioral privacy-technical protection and 3 items for attitude towards BBS. For our later analysis, we did not consider data related to perceived enjoyment. All the respective items (questions) in the constructs are provided as Supplemental Files. We measured the responses to the items on a 7-scale Likert scale from 1 = strongly disagree to 7 = strongly agree.

### Sample organizations

We recruited participants through the website announcement on the University of Saskatchewan's PAWS homepage and on the social networking site, LinkedIn. Participation was entirely voluntary. The participants had to read and accept the consent form to participate in the study. No real identities and email addresses were collected during the data-gathering phase in the surveys. The consent for participation was obtained via an implied consent form. By completing and submitting the questionnaire, participants' free and informed consent was implied and indicated that they understood the conditions of participation in the study spelled out in the consent form.

To contextualize the surveys for SCS, we provided participants at the beginning of the pre-test survey questionnaire (presented as the Article S1) with a video about a brief description of blockchain technology and BBS. The inclusion criteria for the SCS survey was that any individual with knowledge about the internet could participate. After

participants completed the pre-test survey, we presented them with another video about using the SCS and hosted a remote session allowing them to use the SCS for fifteen minutes. We did not record but noted down their comments and confusion during their interaction with the system. Thereafter, we presented them with a post-test survey questionnaire (presented as a Article S2) to measure different constructs of our Augmented TAM with Trust model.

Similarly, we conducted the pre-test and post-test surveys for the DSS part as well. The post-test survey questionnaire for DSS is presented as a Article S3. Each participant in the DSS survey was also asked to use the DSS remotely for fifteen minutes. The inclusion criteria for the DSS survey was that the participants should be from a technical (computer science or engineering) background because the DSS includes technical aspects that only the software developer or system administrator could understand better. Most of the participants completing DSS surveys also took part in the SCS surveys.

### Participants demographics

A total of 66 participants took part in the SCS study and 53 participated in the DSS study. However, upon cleaning, 63 valid responses for SCS and 50 for DSS were left for the analysis. We used a partial least square nonparametric bootstrapping procedure to test the statistical significance with 5,000 subsamples (*Hair et al., 2013*) so that the resampling process would create subsamples with observations randomly drawn from the original set of data. For the study, we based our survey by collecting data from the participants who understood at least something about the blockchain and smart contract technologies after watching the video that we prepared on blockchain technology and BBS. The mean score suggests that for SCS, 79% of participants have basic knowledge and 19% have advanced knowledge of blockchain technology; whereas for DSS, 68% of participants have basic knowledge and 28% have advanced knowledge of blockchain technology. Table 1 highlights the demographics of the participants.

## RESULTS

We used SPSS version 26 to process the collected data with descriptive statistics. We analyzed the research model with structural equation modelling using smartPLS (Partial Least Squares). PLS is a well-established technique for estimating path coefficients in structural models and has been widely used in research studies to model latent constructs under conditions of non-normality and small to medium sample sizes (*Wong, 2013*). The structural equation model (SEM) as suggested by *Hair et al. (2013)* includes the testing of the measurement models (exploratory factor analysis, internal consistency, convergent validity, divergent validity, Dillon-Goldstein's rho) and the structural models (regression analysis). We started by fitting the measurement models to the data and later we tested the underlying structural models.

We applied the path weighting structural model scheme in smartPLS (*Wong, 2013*), which provides the highest $R^2$ value for endogenous or dependent latent variables. The purpose of PLS regression is to combine features from principal component analysis (PCA) and multiple regression (*Roca, García & de la Vega, 2009*). PLS-SEM is applicable

**Table 1 Demographics of participants.**

| Criterion | Subgroup | Number (#) | | Percentage (%) | |
|---|---|---|---|---|---|
| | | SCS | DSS | SCS | DSS |
| **Gender** | Female | 15 | 11 | 24 | 22 |
| | Male | 48 | 39 | 76 | 78 |
| | Other | 0 | 0 | 0 | 0 |
| **Age** | 18–24 | 6 | 4 | 9.5 | 8 |
| | 25–34 | 45 | 34 | 71.4 | 68 |
| | 35–44 | 11 | 9 | 17 | 18 |
| | 44–54 | 1 | 3 | 1.6 | 6 |
| **Highest education completed** | High school | 1 | 0 | 1.6 | 0 |
| | Bachelors | 13 | 12 | 20.6 | 24 |
| | Masters | 39 | 31 | 61.9 | 62 |
| | PhD | 10 | 7 | 15.9 | 14 |
| **Area** | Business | 2 | 1 | 3.2 | 2 |
| | Comp Sc | 39 | 36 | 61.9 | 72 |
| | Engineering | 20 | 13 | 31.7 | 26 |
| | Social Sc | 2 | 0 | 3.3 | 0 |
| **Continent** | Africa | 3 | 2 | 4.8 | 4 |
| | Asia | 30 | 20 | 47.6 | 40 |
| | Europe | 5 | 3 | 7.9 | 6 |
| | N. America | 19 | 18 | 30.2 | 36 |
| | Oceania | 6 | 7 | 9.5 | 14 |
| | S. America | 0 | 0 | 0 | 0 |
| **Familiarity with blockchain and smart contracts** | Extremely | 12 | 14 | 19 | 28 |
| | Moderately | 20 | 19 | 31.7 | 38 |
| | Slightly | 30 | 15 | 47.6 | 30 |
| | Neither | 0 | 1 | 0 | 2 |
| | Slightly Not | 1 | 0 | 1.6 | 0 |
| | Moderately Not | 0 | 0 | 0 | 0 |
| | Extremely Not | 0 | 1 | 0 | 2 |

for all kinds of PLS path model specifications and estimations. We first used 300 maximum iterations for calculating the PLS results and 7 stop criterion values (*Hair et al., 2013*) so that the PLS algorithm could stop when the change in the outer weights between two consecutive iterations was smaller than 7 stop criterion value. We then used a nonparametric bootstrapping procedure to test the statistical significance of various PLS-SEM results that include path coefficients and $R^2$ values. Bootstrapping is a resampling technique with replacement from the sample data to generate empirical sampling distribution. In our case, we used 5,000 subsamples and a two-tailed test type with a 0.1 significance level (*Hair et al., 2013*).

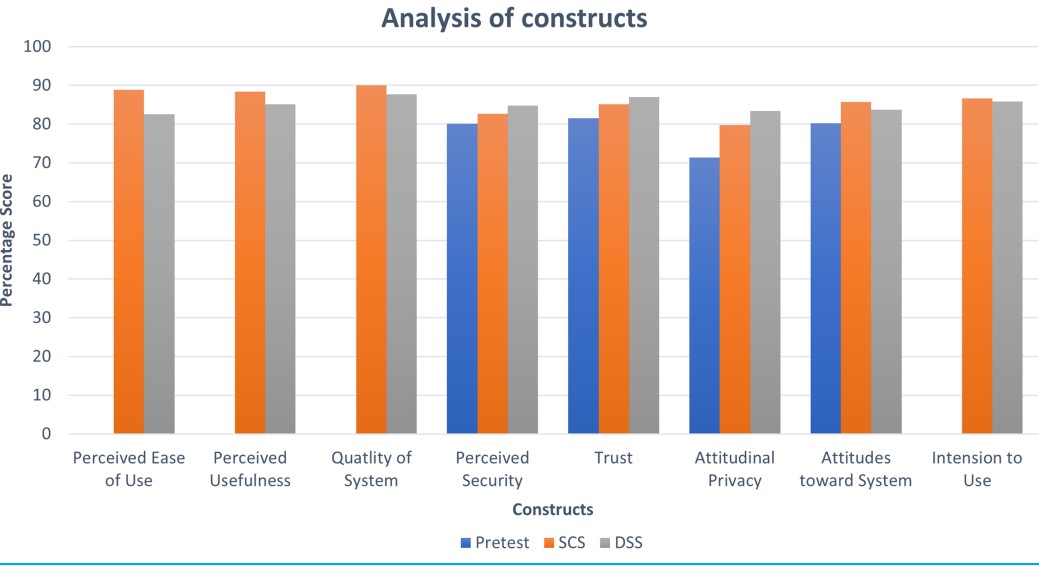

**Figure 4 Analysis of constructs.**

## Descriptive statistic

We had a 7-scale Likert scale for the responses to the items, so we categorized the scale in terms of percentage value to analyze the average score for each item and overall impression of the constructs. We collected scores for all the items in perceived ease of use, perceived usefulness, quality of system, trust, security, privacy, attitudes, and intention to use constructs of our model.

The scores obtained for selected constructs indicate that user perceptions on the benefits of using BBS should be maintained by making improvements to achieve a higher level of score category. The preliminary descriptive statistic of the obtained data is shown in Fig. 4 which informs that the average results of the constructs are above 71.43%, so they qualified for the quite high category (*Shrestha & Vassileva, 2019a*). The comparatively lower pre-test scores indicate that participants developed confidence and trust towards the overall usefulness, usability, attitudes and intention to use the BBS after they used the SCS and DSS. Furthermore, higher scores for PEOS, PU, QOS for SCS over DSS signify that the participants feel easier to use SCS compared to the participants who participated in the DSS part of the study. However, all the selected constructs in our study provided a significant impression in the context of both BBS.

## Measurement validation

We checked the measurement model with the exploratory factor analysis by testing the convergent validity, reliability of measures and discriminant validity.

For Exploratory Factor Analysis, we first checked the factor loadings of individual items, as shown in Table 2, to see whether the items in each variable loaded highly on its own construct over the other respective constructs. According to *Chin, Peterson & Brown (2008)*, factor loadings exceeding 0.60 can be considered as significant. In our study, all the indicators in the measurement models had a factor loading of value greater than 0.60

**Table 2 Exploratory factor analysis.**

| Construct | Item | Factor loading | | |
|---|---|---|---|---|
| | | Pretest | SCS | DSS |
| Attitudinal privacy | AP1 | 0.835 | 0.812 | 0.864 |
| | AP2 | 0.808 | 0.862 | 0.864 |
| | AP3 | 0.88 | 0.774 | 0.685 |
| | AP4 | 0.854 | 0.842 | 0.816 |
| Attitudes towards system | ATS1 | 0.964 | 0.938 | 0.948 |
| | ATS2 | 0.971 | 0.955 | 0.943 |
| | ATS3 | 0.962 | 0.942 | 0.954 |
| Beh privacy-general caution | BP-GC1 | 0.775 | | |
| | BP-GC2 | 0.92 | | |
| | BP-GC3 | 0.888 | | |
| | BP-GC4 | 0.882 | | |
| Beh privacy-technical protection | BP-TP1 | 0.605 | | |
| | BP-TP2 | 0.869 | | |
| | BP-TP3 | 0.775 | | |
| | BP-TP4 | 0.39 | | |
| Security | PS1 | 0.919 | 0.891 | 0.899 |
| | PS2 | 0.922 | 0.858 | 0.934 |
| | PS3 | 0.883 | 0.859 | 0.893 |
| Trust | T1 | 0.854 | 0.821 | 0.88 |
| | T2 | 0.877 | 0.792 | 0.899 |
| | T3 | 0.858 | 0.721 | 0.809 |
| | T4 | 0.82 | 0.765 | 0.82 |
| | T5 | 0.766 | 0.753 | 0.783 |
| | T6 | 0.727 | 0.786 | 0.8 |
| | T7 | 0.735 | 0.696 | 0.734 |
| | T8 | 0.924 | 0.806 | 0.817 |
| | T9 | 0.673 | 0.819 | 0.931 |
| Intention to use | ITU1 | | 0.89 | 0.804 |
| | ITU2 | | 0.917 | 0.936 |
| | ITU3 | | 0.854 | 0.926 |
| Perceived ease of use | PEOU1 | | 0.817 | 0.885 |
| | PEOU2 | | 0.818 | 0.888 |
| | PEOU3 | | 0.798 | 0.883 |
| | PEOU4 | | 0.805 | 0.809 |
| | PEOU5 | | 0.866 | 0.755 |
| | PEOU6 | | 0.876 | 0.865 |
| Perceived usefulness | PU1 | | 0.743 | 0.857 |
| | PU2 | | 0.685 | 0.843 |
| | PU3 | | 0.661 | 0.883 |
| | PU4 | | 0.856 | 0.871 |

| Construct | Item | Factor loading | | |
|---|---|---|---|---|
| | | Pretest | SCS | DSS |
| | PU5 | | 0.725 | 0.811 |
| | PU6 | | 0.73 | 0.749 |
| Quality of system | QOS1 | | 0.88 | 0.896 |
| | QOS2 | | 0.901 | 0.943 |
| | QOS3 | | 0.83 | 0.872 |
| | QOS4 | | 0.71 | 0.868 |

**Table 3 Constructs reliability and validity.**

| Construct | Pretest | | | SCS | | | DSS | | |
|---|---|---|---|---|---|---|---|---|---|
| | rho_A | CR | AVE | rho_A | CR | AVE | rho_A | CR | AVE |
| Attitudes towards system | 0.967 | 0.976 | 0.932 | 0.94 | 0.962 | 0.893 | 0.944 | 0.964 | 0.899 |
| Intention to use | X | X | X | 0.873 | 0.917 | 0.787 | 0.875 | 0.92 | 0.794 |
| Perceived ease of use | X | X | X | 0.923 | 0.93 | 0.69 | 0.928 | 0.939 | 0.721 |
| Perceived usefulness | X | X | X | 0.834 | 0.876 | 0.541 | 0.917 | 0.933 | 0.7 |
| Atd privacy or privacy | 0.875 | 0.909 | 0.714 | 0.843 | 0.894 | 0.678 | 0.838 | 0.884 | 0.657 |
| Quality of system | X | X | X | 0.872 | 0.9 | 0.695 | 0.928 | 0.942 | 0.801 |
| Security | 0.894 | 0.934 | 0.825 | 0.84 | 0.903 | 0.756 | 0.895 | 0.935 | 0.826 |
| Trust | 0.942 | 0.944 | 0.652 | 0.919 | 0.931 | 0.599 | 0.948 | 0.953 | 0.693 |
| Beh privacy-general caution | 0.93 | 0.924 | 0.753 | X | X | X | X | X | X |
| Beh privacy-technical protection | 0.285 | 0.766 | 0.469 | X | X | X | X | X | X |

except for Item 4 in the construct Behavioral Privacy-Technical Protection (BP-TP4). Since the square of factor loading is directly translated as item's reliability, the item BP-TP4, "I regularly clear my browser's history" with a very low loading value of 0.39 indicated that its communality value would be only 0.15, and thus should be avoided in the model. Although we used the validated constructs, our exploratory analysis detected that the item BP-TP4 had a weak influence on the Behavioral Privacy construct.

For the Convergent Validity of each construct measure, we calculated the Average Variance Extracted (AVE) and Composite Reliability (CR) from the factor loading. AVE for each construct should exceed the recommended level of 0.50 so that over 50% of the variances observed in the items were accounted for by the hypothesized constructs, and CR should also be above 0.75 to publish results (*Hair et al., 2014*). In our study, the AVE reported in Table 3 exceeds 0.50 for all the constructs except for Beh Privacy-Technical Protection (BP-TP). However, CR for each construct was above 0.75 (acceptable), confirming that it measures the construct validity of the model. Since the BP-TP had the item BP-TP4 of very low factor loading along with an AVE value of 0.469, it suggests that the factor BP-TP did not bring significant variance for the variables (items/questions) to converge into a single construct which means BP-TP items are a less-than-effective

measure of the latent construct. We also justify this with the exceptionally low rho_A value for the construct BP-TP.

Table 3 shows the calculated rho_A value (Dillon-Goldstein's rho) for checking the internal consistency to justify the reliability of each measure. The rho_A evaluates the within-scale consistency of the responses to the items of the measures of constructs and is a better reliability measure than Cronbach's alpha in SEM (*Demo et al., 2012*). In our study, as recommended, rho_A for each construct was greater than 0.70 except for BP-TP which had a 0.28 rho value. Therefore, this also supports our decision of removing the behavioral privacy constructs from the post-tests for both SCS and DSS. We assumed that using the BBS simply does not influence the user's behavioral perception of privacy. So, we were interested to see if there is any significant effect on the attitudinal aspect of privacy.

For assessing the Discriminant Validity of measures, we calculated the square root of the AVE (along the diagonals) of each construct as shown in Table 4. To lean towards discriminant validity (*Fornell & Larcker, 1981*) recommended having low correlations between the measure of interest and the measures of other constructs. In our model, we observed those diagonal values for each construct exceeded other corresponding values, which are the intercorrelations of the given construct with the other remaining constructs. This pointed out that the measures of each construct which was theoretically supposed to be not overlapping with measures of other variables are in fact, unrelated in our model.

## Partial least square path modeling

To begin our Structural Equation Modeling (SEM) analysis, we built the models for the general population in the context of the pre-test (prototype model) and two subsystems SCS and DSC. We characterized the models by looking into coefficients of determination ($R^2$'s), path coefficients ($\beta$'s) and corresponding *P*-value. $R^2$ determines the variance of a given construct explained by antecedents, $\beta$ captures the strength of the relationship between the selected constructs and P-value determines the statistical significance of the models (*Shrestha & Vassileva, 2019a*). According to Chin's guideline (*Chin, Marcelin & Newsted, 2003*), a path coefficient should be equal to or greater than 0.2 to be considered relevant. A model is statistically somewhat significant (*p) when *p*-value < 0.1, statistically quite significant (**p) when *p*-value < 0.01 and statistically highly significant (***p) when *p*-value < 0.001. Tables 5–7 each show the standardized path coefficient ($\beta$), t-statistics, *p*-value and $R^2$ across selected constructs for pre-test, SCS and DSS, respectively. The indirect and total effects of one construct over another construct in the presence of mediating constructs were also computed alongside.

## Validation of hypotheses

For pre-test in the context of prototype model, the model presented in Fig. 5 shows causal relationship between perceived attitudinal privacy, behavioral privacy-technical protection, behavioral privacy-general caution, perceived security, trust and attitude towards BBS constructs. Considering the direct effects, attitudinal privacy (privacy concern) had very high significant effects on security ($\beta = 0.64$; $p < 0.001$) and trust

**Table 4 Discriminant validity.**

**Pretest**

| Construct | AP | ATS | | BP-GC | BP-TP | S | T |
|---|---|---|---|---|---|---|---|
| Atd Privacy | 0.845 | | | | | | |
| Attitudes toward BSS | 0.58 | 0.965 | | | | | |
| Beh privacy-general caution | 0.465 | 0.285 | | 0.868 | | | |
| Beh privacy-technical protection | 0.068 | 0.187 | | 0.375 | 0.684 | | |
| Security | 0.637 | 0.535 | | 0.303 | −0.065 | 0.908 | |
| Trust | 0.65 | 0.762 | | 0.275 | 0.162 | 0.728 | 0.808 |

**Shopping Cart System (SCS)**

| | ATS | ITU | PEOU | PU | P | QOS | S | T |
|---|---|---|---|---|---|---|---|---|
| Attitudes toward SCS | 0.945 | | | | | | | |
| Intention to use | 0.61 | 0.887 | | | | | | |
| Perceived ease of use | 0.557 | 0.543 | 0.831 | | | | | |
| Perceived usefulness | 0.553 | 0.691 | 0.615 | 0.736 | | | | |
| Privacy | 0.617 | 0.587 | 0.482 | 0.489 | 0.823 | | | |
| Quality of system | 0.509 | 0.69 | 0.508 | 0.691 | 0.299 | 0.834 | | |
| Security | 0.374 | 0.398 | 0.233 | 0.344 | 0.654 | 0.282 | 0.87 | |
| Trust | 0.677 | 0.653 | 0.599 | 0.56 | 0.748 | 0.395 | 0.61 | 0.774 |

**Data Sharing System (DCS)**

| | ATS | ITU | PEOU | PU | P | QOS | S | T |
|---|---|---|---|---|---|---|---|---|
| Attitude towards DSS | 0.948 | | | | | | | |
| Intension to use | 0.766 | 0.891 | | | | | | |
| Perceived ease of use | 0.661 | 0.573 | 0.849 | | | | | |
| Perceived usefulness | 0.672 | 0.725 | 0.782 | 0.837 | | | | |
| Privacy | 0.596 | 0.496 | 0.666 | 0.656 | 0.81 | | | |
| Quality of system | 0.688 | 0.631 | 0.685 | 0.762 | 0.708 | 0.895 | | |
| Security | 0.563 | 0.447 | 0.592 | 0.68 | 0.824 | 0.7 | 0.909 | |
| Trust | 0.705 | 0.557 | 0.718 | 0.713 | 0.8 | 0.775 | 0.777 | 0.832 |

($\beta = 0.313$; $p < 0.001$), but an insignificant effect on attitudes towards system ($\beta = 0.176$; $p > 0.05$). In addition, attitudinal privacy also positively affected behavioral privacy-general caution ($\beta = 0.465$; $p < 0.001$) but had an insignificant effect on behavioral privacy-technical protection ($\beta = 0.068$; $p > 0.1$). The effect of security on trust was also highly significant ($\beta = 0.529$; $p < 0.001$), but insignificant on attitudes towards BBS ($\beta = -0.104$; $p > 0.1$). Finally, trust had a high significant positive effect on attitudes towards BBS ($\beta = 0.724$; $p < 0.001$). Thus, hypotheses H7, H8, H10, H11 and H13 were supported, but H9, H12, and H14 were rejected in the context of pre-test. Moreover, trust, privacy and security explain 59.8% of variance in attitudes towards BBS ($R^2 = 0.598$), security and privacy explain 58.8% of variance in trust ($R^2 = 0.588$), privacy explains 40.6% of variance in security ($R^2 = 0.406$), whereas attitudinal privacy explains very low, 21.6% of variance on behavioral privacy-general caution ($R^2 = 0.216$) and 0.5% on behavioral

**Table 5 Structural estimates (hypotheses testing) for pre-test.**

| Structural path | Direct effect | | | Total effect | | | Indirect effect | | | |
|---|---|---|---|---|---|---|---|---|---|---|
| | Std β | T | P | Std β | T | P | Std β | T | P | VAF |
| Atd privacy → Attitudes towards system | 0.176 | 1.4 | 0.162 | 0.586 | 6.403 | 0 | 0.412 | 4.466 | 0 | 0.703 |
| Atd privacy → Beh privacy-general caution | 0.465 | 5.123 | 0 | 0.474 | 5.123 | 0 | | | | |
| Atd privacy → Beh privacy-tech protection | 0.068 | 0.366 | 0.715 | 0.036 | 0.366 | 0.715 | | | | |
| Atd privacy → Security | 0.64 | 9.94 | 0 | 0.64 | 9.94 | 0 | | | | |
| Atd privacy → Trust | 0.313 | 3.8 | 0 | 0.654 | 10.921 | 0 | 0.341 | 5.301 | 0 | 0.521 |
| Security → Attitudes towards system | −0.104 | 0.865 | 0.387 | 0.283 | 2.396 | 0.017 | 0.391 | 4.348 | 0 | 1.382 |
| Security → Trust | 0.529 | 6.446 | 0 | 0.529 | 6.446 | 0 | | | | |
| Trust → Attitudes towards system | 0.724 | 6.33 | 0 | 0.732 | 6.33 | 0 | | | | |

Notes:
Direct effect column is when all latent variables are present in the model without any exclusion.
$R^2$ (Attitude=0.598; BP-GC=0.216; BP-TP=0.005; Security=0.406; Trust=0.588).

**Table 6 Structural estimates (hypotheses testing) for SCS.**

| Structural path | Direct effect | | | Total effect | | | Indirect effect | | | |
|---|---|---|---|---|---|---|---|---|---|---|
| | Std β | T | P | Std β | T | P | Std β | T | P | VAF |
| Attitude towards system → Intention to use | 0.25 | 1.713 | 0.087 | 0.25 | 1.713 | 0.087 | | | | |
| Perceived ease of use → Intention to use | 0.058 | 0.449 | 0.653 | 0.153 | 1.201 | 0.23 | 0.108 | 1.363 | 0.173 | 0.706 |
| Perceived ease of use → Perceived usefulness | 0.356 | 2.902 | 0.004 | 0.362 | 2.902 | 0.004 | | | | |
| Perceived usefulness → Intention to use | 0.284 | 1.743 | 0.081 | 0.297 | 1.743 | 0.081 | | | | |
| Privacy → Attitude towards SCS | 0.325 | 2.828 | 0.005 | 0.62 | 7.626 | 0 | 0.293 | 2.23 | 0.026 | 0.473 |
| Privacy → Security | 0.654 | 8.333 | 0 | 0.664 | 8.333 | 0 | | | | |
| Privacy → Trust | 0.597 | 4.58 | 0 | 0.752 | 11.855 | 0 | 0.155 | 1.225 | 0.221 | 0.206 |
| Quality of system → Intention to use | 0.338 | 2.116 | 0.034 | 0.49 | 4.116 | 0 | 0.152 | 1.557 | 0.12 | 0.31 |
| Quality of system → Perceived usefulness | 0.509 | 4.731 | 0 | 0.509 | 4.731 | 0 | | | | |
| Security → Attitude towards SCS | −0.165 | 1.536 | 0.125 | −0.049 | 0.349 | 0.727 | 0.122 | 1.191 | 0.234 | -2.489 |
| Security → Trust | 0.212 | 1.339 | 0.181 | 0.229 | 1.339 | 0.181 | | | | |
| Trust → Attitude towards SCS | 0.534 | 3.349 | 0.001 | 0.538 | 3.349 | 0.001 | | | | |

Notes:
Direct effect column is when all latent variables are present in the model without any exclusion.
$R^2$ (Attitude= 0.5; Intention=0.612; Perceived Usefulness=0.571; Security=0.428; Trust=0.585).

privacy-technical protection. $R^2$ value higher than 0.26 indicates a substantial model (*Muller & Cohen, 1989*).

For post-test study in the context of SCS, the model presented in Fig. 6 shows causal relationship between perceived ease of use, perceived usefulness, quality of system, security, privacy, trust, attitude towards SCS and intention to use SCS constructs. Considering the direct effect, perceived ease of use had quite significant effect on perceived usefulness (β = 0.356; $p < 0.01$) but insignificant effect on intention to use (β = 0.058; $p > 0.1$); therefore, H1 was supported and H2 was rejected. Perceived usefulness had relevant but somewhat significant effect on intention to use (β = 0.284; $p < 0.1$); thus, H3 was also supported. Quality of system had positive significant effect on perceived

**Table 7 Structural estimates (hypotheses testing) for DSS.**

| Structural path | Direct effect | | | Total effect | | | Indirect effect | | | |
|---|---|---|---|---|---|---|---|---|---|---|
| | Std β | T | P | Std β | T | P | Std β | T | P | VAF |
| Attitude towards system → Intention to use | 0.554 | 3.967 | 0 | 0.554 | 3.967 | 0 | | | | |
| Perceived ease of use → Intention to use | −0.173 | 0.979 | 0.327 | 0.069 | 0.5 | 0.617 | 0.242 | 1.946 | 0.052 | 3.507 |
| Perceived ease of use → Perceived usefulness | 0.488 | 3.456 | 0.001 | 0.489 | 3.456 | 0.001 | | | | |
| Perceived usefulness → Intention to use | 0.495 | 2.354 | 0.019 | 0.495 | 2.354 | 0.019 | | | | |
| Privacy → Attitude towards system | 0.097 | 0.383 | 0.701 | 0.596 | 5.531 | 0 | 0.5 | 2.319 | 0.02 | 0.839 |
| Privacy → Security | 0.82 | 15.161 | 0 | 0.824 | 15.161 | 0 | | | | |
| Privacy → Trust | 0.495 | 3.54 | 0 | 0.8 | 14.958 | 0 | 0.304 | 2.59 | 0.01 | 0.38 |
| Quality of system → Intention to use | −0.009 | 0.046 | 0.964 | 0.202 | 1.063 | 0.288 | 0.212 | 1.846 | 0.065 | 1.05 |
| Quality of system → Perceived usefulness | 0.427 | 2.667 | 0.008 | 0.427 | 2.667 | 0.008 | | | | |
| Security → Attitude towards system | −0.012 | 0.053 | 0.958 | 0.223 | 0.998 | 0.318 | 0.235 | 2.113 | 0.035 | 1.05 |
| Security → Trust | 0.369 | 2.566 | 0.01 | 0.369 | 2.566 | 0.01 | | | | |
| Trust → Attitude towards system | 0.637 | 4.316 | 0 | 0.637 | 4.316 | 0 | | | | |

Notes:
Direct effect column is when all latent variables are present in the model without any exclusion.
$R^2$ (Attitude=0.5; Intension to use=0.678; Perceived Usefulness=0.708; Security=0.679; Trust=0.683).

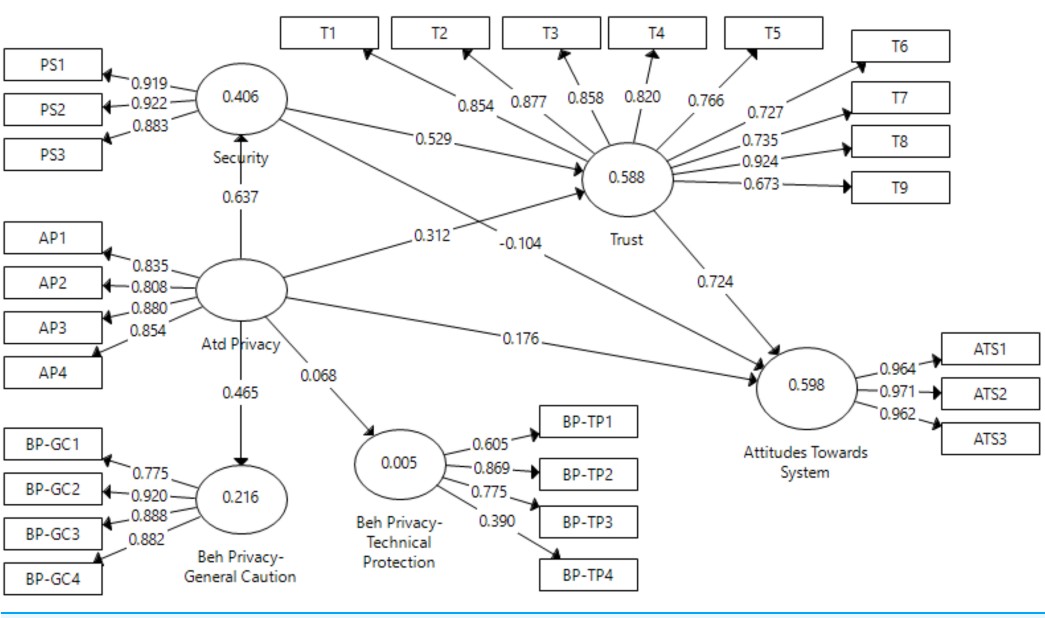

**Figure 5 Pretest direct effect.**

usefulness (β = 0.509; *p* < 0.001) and somewhat significant effect on intention to use SCS (β = 0.338; *p* < 0.1); therefore, H4 and H5 were supported. Attitude towards SCS had relevant but somewhat significant effect on intention to use (β = 0.25; *p* < 0.1); therefore, H6 was supported. The effect of trust was highly significant on attitude towards SCS (β = 0.534; *p* < 0.001); therefore, H7 was supported. Perceived privacy had positive significant effects on trust (β = 0.609; *p* < 0.001), attitudes towards SCS (β = 0.325; *p* < 0.01) and perceived security (β = 0.654; *p* < 0.001); therefore, H8, H9 and H10 were supported.

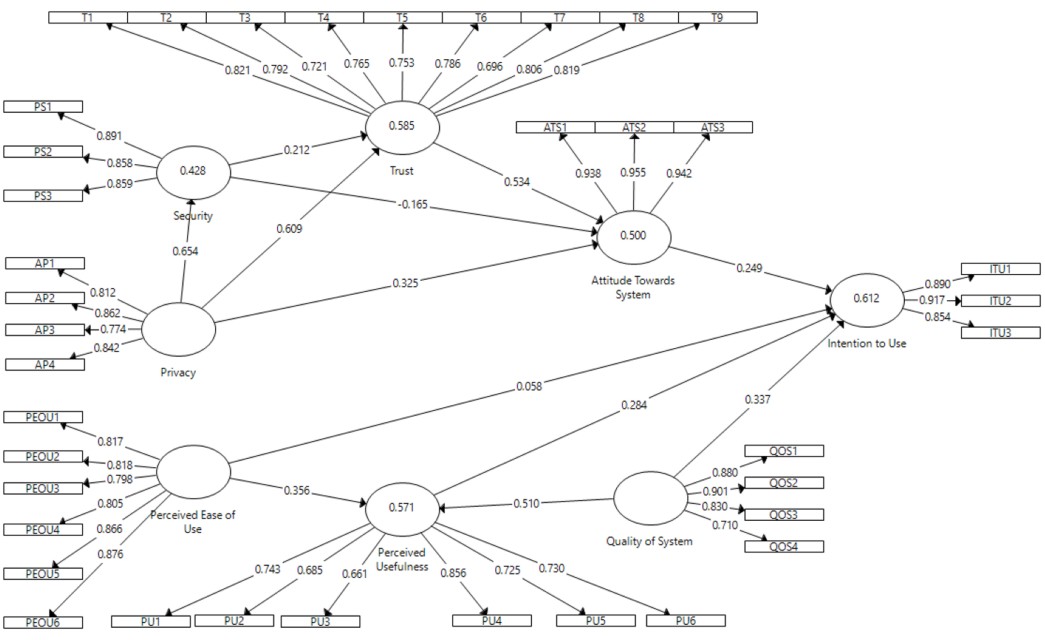

**Figure 6 Shopping Cart System (SCS) direct effect.**

Perceived security had insignificant effect on trust ($\beta = 0.212$; $p > 0.1$) and attitudes towards SCS ($\beta = -0.165$; $p > 0.1$); therefore, H13 and H14 were rejected. In the following, the explained variances include perceived usefulness ($R^2 = 0.571$), security ($R^2 = 0.428$), trust ($R^2 = 0.585$), attitude towards SCS ($R^2 = 0.5$) and intention to use ($R^2 = 0.612$). Therefore, $R^2$ value higher than 0.26 indicated a substantial model for SCS (*Muller & Cohen, 1989*).

Similarly, for post-test study in the context of DSS, the model presented in Fig. 7 shows causal relationship between perceived ease of use, perceived usefulness, quality of system, security, privacy, trust, attitude towards DSS and intention to use DSS constructs. Considering the direct effect, perceived ease of use had significant effect on perceived usefulness ($\beta = 0.488$; $p < 0.001$) but insignificant effect on intention to use ($\beta = -0.173$; $p > 0.1$); therefore, H1 was supported and H2 was rejected. Perceived usefulness had relevant but somewhat significant effect on intention to use ($\beta = 0.495$; $p < 0.1$); thus, H3 was also supported. Quality of system had positive significant effect on perceived usefulness ($\beta = 0.427$; $p < 0.01$), but insignificant effect on intention to use DSS ($\beta = -0.009$; $p > 0.1$); therefore, H4 was supported and H5 was rejected. Attitude towards DSS had relevant and positive significant effect on intention to use ($\beta = 0.554$; $p < 0.001$); therefore, H6 was supported. The effect of trust was highly significant on attitude towards DSS ($\beta = 0.637$; $p < 0.001$); therefore, H7 was supported. Perceived privacy had positive significant effects on trust ($\beta = 0.495$; $p < 0.001$) and perceived security ($\beta = 0.82$; $p < 0.001$), but insignificant effect on attitudes towards DSS ($\beta = 0.097$; $p > 0.1$); therefore, H8 and H10 were supported but H9 was rejected. Perceived security had significant effect on trust ($\beta = 0.369$; $p < 0.01$) but insignificant effect on attitudes towards DSS ($\beta = -0.012$; $p > 0.1$); therefore, H13 was supported but H14 was rejected. In the following, the explained variances

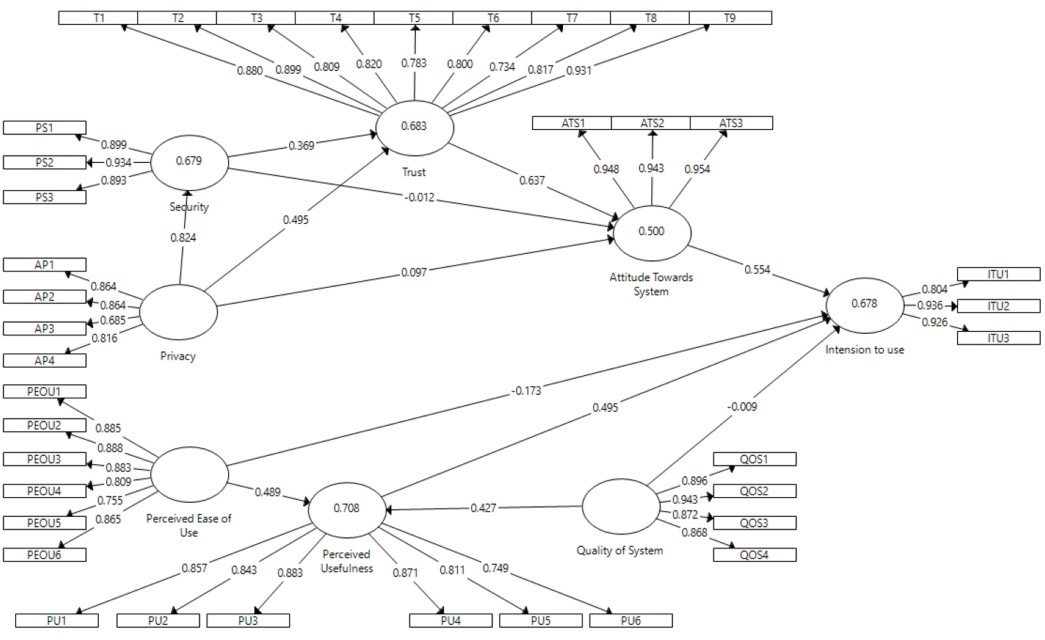

**Figure 7 Data Sharing System (DSS) direct effect.**

include perceived usefulness ($R^2 = 0.708$), security ($R^2 = 0.679$), trust ($R^2 = 0.683$), attitude towards SCS ($R^2 = 0.5$) and intention to use ($R^2 = 0.678$). Therefore, $R^2$ value higher than 0.26 indicated a substantial model for DSS (*Muller & Cohen, 1989*). Table 8 summarizes the validation of our study's hypotheses.

## Total effect analysis

To address the first research question, we present the total effect of antecedents from the trust model on attitudes towards BBS as shown in Fig. 8. In the pre-test model, trust had the strongest total effect on attitudes towards BBS ($\beta = 0.732$; $p < 0.001$), followed by privacy on attitudes towards BBS ($\beta = 0.586$; $p < 0.001$) and security on attitudes towards BBS ($\beta = 0.283$; $p < 0.1$), which was marginally significant. In the SCS model, privacy had the strongest influence on attitudes towards SCS ($\beta = 0.62$; $p < 0.001$), followed by trust on attitudes towards SCS ($\beta = 0.538$; $p < 0.001$), while security had no significant total effect on attitudes towards SCS ($\beta = -0.049$; $p > 0.1$). Finally, the total effect statistic for the DSS model was similar to that of the pre-test model, with respect to first two strongest design constructs which were trust ($\beta = 0.637$; $p < 0.001$), followed by privacy ($\beta = 0.596$; $p < 0.001$). Security turned out to have no significant effect on attitude towards DSS ($\beta = 0.223$; $p > 0.1$).

To address the second research question, we present the total effect of the perceived design constructs on intention to use from SCS and DSS model as shown in Fig. 9. In SCS model, quality of system had the strongest total effect on intention to use SCS ($\beta = 0.49$; $p < 0.001$). Perceived usefulness had a weak total effect on intention to use SCS ($\beta = 0.297$; $p < 0.1$), while privacy had no significant total effect on intention to use SCS ($\beta = 0.156$;

**Table 8 Validation of the study's hypotheses.**

| | Hypothesis | Pre-test | SCS | DDS |
|---|---|---|---|---|
| H1 | Perceived ease of use significantly influences perceived usefulness of BBS. | | √ | √ |
| H2 | Perceived ease of use significantly influences intention to use BBS. | | × (× → Usefulness) | × (√ → Usefulness) |
| H3 | Perceived usefulness significantly influences intention to use BBS. | | √ | √ |
| H4 | Quality of system significantly influences perceived usefulness of BBS. | | √ | √ |
| H5 | Quality of system significantly influences intention to use BBS. | | √ (× → Usefulness) | × (√ → Usefulness) |
| H6 | Attitude towards BBS significantly influences intention to use BBS. | | √ | √ |
| H7 | Trust positively affects users' attitudes toward BBS. | √ | √ | √ |
| H8 | Perceived privacy has a positive effect on the users' trust in BBS. | √ | √ | √ |
| H9 | Perceived privacy has a positive effect on the users' attitudes toward BBS. | × (√ → Trust) (× → Security) | √ (√ → Trust) (× → Security) | × (√ → Trust) (× → Security) |
| H10 | Perceived privacy positively or negatively affects users' perceived security. | √ | √ | √ |
| H11 | Privacy concern positively affects users' behavior on general caution. | √ | | |
| H12 | Privacy concern positively affects users' behavior on technical protection. | × | | |
| H13 | Perceived security positively affects users' trust in BBS. | √ | × | √ |
| H14 | Perceived security positively affects users' attitudes toward BBS. | × (√ → Trust) | × (× → Trust) | × (× → Trust) |

**Note:**
√ = True; × = False; √-> = Mediated by latent variable; ×-> = Not mediated by latent variable.

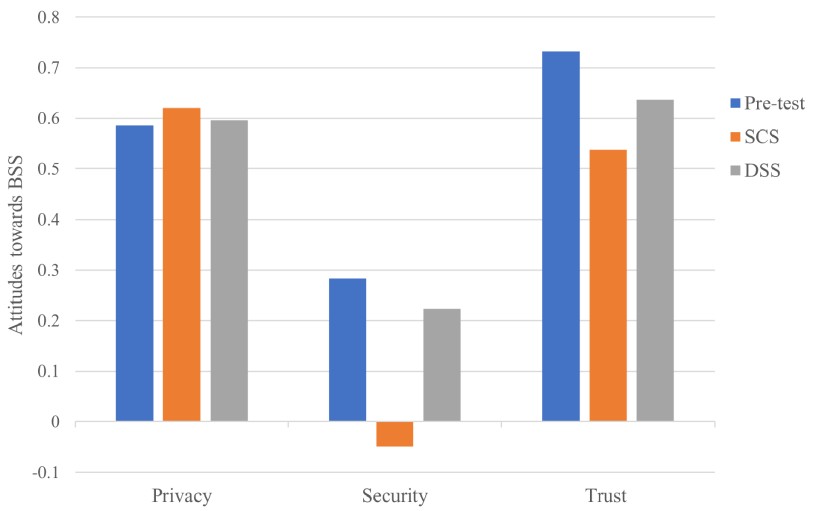

**Figure 8 Total effect of the trust design constructs on attitudes towards BBS.**

$p > 0.1$), followed by perceived ease of use on intention to use SCS ($\beta = 0.153$; $p > 0.1$) and security on intention to use SCS ($\beta = -0.01$; $p > 0.1$). In the context of DSS, Perceived usefulness had the strongest total effect on intention to use DSS ($\beta = 0.495$; $p < 0.01$), followed by trust on intention to use DSS ($\beta = 0.353$; $p < 0.001$) and privacy on intention to use DSS ($\beta = 0.33$; $p < 0.001$). Quality of system had no significant total effect on intention to use DSS ($\beta = 0.202$; $p > 0.1$), followed by security on intention to use DSS ($\beta = 0.124$; $P > 0.1$) and perceived ease of use on intention to use DSS ($\beta = 0.069$; $p > 0.1$).

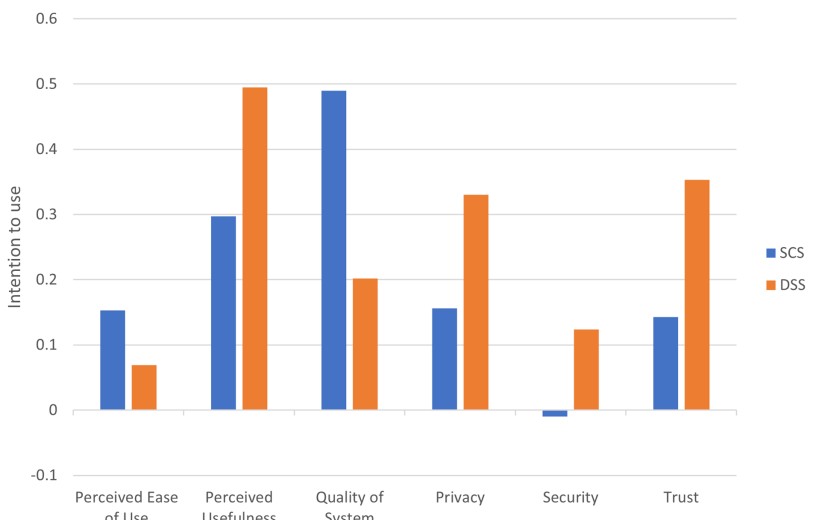

**Figure 9 Total effect of predictors on intention to use.**

## Mediation analysis

To address our third, fourth and fifth research questions, we carried out the indirect effect analysis. We first investigated the mediating effect of security and/or trust over the relationship between privacy and attitudes towards BBS, then investigated the mediating effect of trust over the relationship between security and attitudes towards BBS, and finally investigated a similar mediating effect of perceived usefulness over the relationship between ease of use/quality of system on intention to use BBS. According to *Baron & Kenny (1986)*, *Hair et al. (2014)*, there is no need to check for the indirect effect if the direct effect is insignificant in the model.

So, for the pre-test model as presented in Table 5, we found the observed indirect effects for the selected predictors in the presence of mediating variables in the pre-test model. In the presence of mediating effect of both trust and security, the effect of privacy on attitude towards BBS slightly decreased from ($\beta$ = 0.584; $T$ = 6.868; $p$ < 0.001 while excluding both trust and security) to ($\beta$ = 0.412; $T$ = 4.466; $p$ < 0.001) with the *variance accounted for* (VAF) value of 0.703. The VAF is calculated as the ratio of the indirect path coefficient to the total path coefficient. With 70.3% VAF, trust and security had a partial mediation effect between privacy and attitude towards BBS. While analyzing the individual mediating effects between privacy and attitudes towards BBS, trust alone had positive significant effect ($\beta$ = 0.226; $T$ = 3.151; $p$ < 0.01), but security alone had no significant effect ($\beta$ = −0.068; $T$ = 0.852; $p$ > 0.1). So, our finding suggested that only trust played a crucial mediating role while security had no significant effect between privacy and attitudes toward BBS. Similarly, in the presence of mediating effect of trust, the effect of security on attitude towards BBS slightly decreased from ($\beta$ = 0.538; $T$ = 6.14; $p$ < 0.001 while excluding Trust) to ($\beta$ = 0.391; $T$ = 4.348; $p$ < 0.001) with the *variance accounted for* (VAF) value of 1.382. With 138% VAF, trust had a perfect mediation effect between security and attitude towards BBS.

For SCS model, as presented in Table 6, in the presence of mediating effect of trust and security, the effect of privacy on attitude towards SCS slightly decreased from ($\beta$ = 0.619; $T$ = 8.504; $p < 0.001$ while excluding both trust and security) to ($\beta$ = 0.293; $T$ = 2.23; $p < 0.1$) with the *variance accounted for* (VAF) value of 0.473. With 47.3% VAF, trust and security had a partial mediation effect between privacy and attitude towards SCS. While analyzing the individual mediating effects between privacy and attitudes towards SCS, trust alone had positive significant effect ($\beta$ = 0.326; $T$ = 2.683; $p < 0.01$), but security alone had no significant effect ($\beta$ = −0.108; $T$ = 1.466; $p > 0.1$). So, our finding suggested that only trust played a crucial mediating role while security had no significant effect between privacy and attitudes toward SCS. In the same SCS model, with the presence of mediating effect of trust, the effect of security on attitude towards SCS became insignificant from ($\beta$ = 0.384; $T$ = 3.304; $p < 0.001$ while excluding trust) to ($\beta$ = 0.122; $T$ = 1.191; $p > 0.1$) with the variance accounted for (VAF) value of −2.489. Therefore, trust had no mediating effect between security and attitude towards SCS since after adding trust predictor as a mediator, the indirect effect on attitude towards SCS became non-significant while the direct effect was also insignificant. Furthermore, in the same SCS model, with the presence of mediating effect of perceived usefulness, the effect of quality of system on intention to use SCS became insignificant from ($\beta$ = 0.705; $T$ = 11.127; $p < 0.001$ while excluding usefulness) to ($\beta$ = 0.152; $T$ = 1.557; $p > 0.1$) with the variance accounted for (VAF) value of 0.31. Therefore, perceived usefulness had no significant mediating effect between quality of system and intention to use SCS since after adding perceived usefulness predictor as a mediator, the indirect effect on intention to use SCS became non-significant while the direct effect was still significant.

In addition, with the presence of mediating effect of perceived usefulness, the ease of use on intention to use SCS became insignificant from ($\beta$ = 0.559; $T$ = 7.035; $p < 0.1$ while excluding usefulness) to ($\beta$ = 0.108; $T$ = 1.363; $p > 0.1$) with the variance accounted for (VAF) value of 0.706. Therefore, perceived usefulness had no significant mediating effect between ease of use and intention to use SCS since after adding perceived usefulness predictor as a mediator, the indirect effect on intention to use SCS became non-significant while the direct effect was also non-significant.

Finally, in the DSS model, as presented in Table 7, in the presence of mediating effect of trust and security, the effect of privacy on attitude towards DSS slightly decreased from ($\beta$ = 0.6; $T$ = 6.134; $p < 0.001$ while excluding both trust and security) to ($\beta$ = 0.5; $T$ = 2.319; $p < 0.1$) with the variance accounted for (VAF) value of 0.839. With 83.9% VAF, trust and security had a partial mediation effect between privacy and attitude towards DSS. While analyzing the individual mediating effects between privacy and attitudes towards DSS, trust alone had positive significant effect ($\beta$ = 0.316; $T$ = 2.726; $p < 0.01$), but security alone had no significant effect ($\beta$ = −0.01; $T$ = 0.053; $p > 0.1$). So, our finding suggested that only trust played a crucial mediating role while security had no significant effect between privacy and attitudes toward DSS. In the same DSS model, no mediation effect was observed for trust between security and attitudes towards DSS. Furthermore, in the same DSS model, with the presence of mediating effect of perceived usefulness, the effect of quality of system on intention to use DSS reduced from ($\beta$ = 0.66; $T$ = 7.82; $p < 0.001$ while

excluding usefulness) to ($\beta = 0.212$; $T = 1.846$; $p < 0.1$) with the variance accounted for (VAF) value of 1.05. Therefore, perceived usefulness had a complete significant mediating effect between quality of system and intention to use DSD since after adding perceived usefulness predictor as a mediator, the indirect effect on intention to use DSD became significant while the direct effect was insignificant. This suggested that the indirect significant path between quality of system and intention to use DSD was contributed by perceived usefulness predictor construct. In addition, with the presence of mediating effect of perceived usefulness, the effect of ease of use on intention to use reduced from ($\beta = 0.588$; $T = 6.385$; $p < 0.001$ while excluding usefulness) to ($\beta = 0.242$; $T = 1.946$; $p < 0.1$) with the variance accounted for (VAF) value of 3.507. Therefore, perceived usefulness had a complete mediating effect between ease of use and intention to use DSD since after adding perceived usefulness predictor as a mediator, the indirect effect on intention to use DSD became significant while the direct effect was non-significant. This suggested that the indirect significant path between perceived ease of use and intention to use DSD was contributed by perceived usefulness predictor construct.

## DISCUSSION

The purpose of this study is to evaluate the user acceptance of a working blockchain-based system by observing the attributes affecting the development of users' attitudes and intention to use the system. We achieved the goal of our research by testing the augmented TAM with a Trust model on our application (BBS) that is built using blockchain technology. The empirical study validates our research model and supports most of the research hypotheses that were set considering the aim of this study. We also identified different issues influencing users' attitudes and intentions to adopt BBS by considering observed facts from the causal relationships and their implications. According to *Gefen, Karahanna & Straub (2003)*, extending TAM with trust model is well justified for its effectiveness in improving the predictive power of the explored issues associated with the acceptance of online services. BBS can be considered as a set of online services, so applying the TAM augmented with trust, as we did in our study, is justified.

The major contribution of our study to the existing literature of blockchain and distributed ledger technologies is to uncover the dimensions and role of trust alongside primary TAM-based design predictors and their causal relationship with users' attitudes and behavioral intention to accept such technologies.

According to the results from our research, TAM-based predictors or trust constructs cannot be applied uniformly to BBS. Depending on the specifics of the BBS, the relationships between perceived trust, perceived security, perceived privacy and attitudes towards the system might change. In this study, there was a customer-specific system: SCS and a company-specific system: DSS. Every participant who completed the post-survey for DSS also completed the post-survey for SCS with 66 participants completing the post-survey for SCS and 53 participants completing the post-survey for DSS. There was no major difference between the users of each system that could lead to the difference between the responses of the two surveys. With the SCS, the user engaged with the point of

view of a customer, whereas the DSS had the user engaged the system as an enterprise's system administrator.

A previous study by *Buchanan et al. (2007)* suggests that attitudinal privacy, in the privacy model, correlates significantly with behavioral privacy-general caution but not significantly with the technical protection factor. The findings of our current research also indicate comparable results. Users who are concerned with their data privacy tend to be more cautious and careful about protecting it, however, if the users are technically competent, they have already used tools to protect their privacy such as clearing the browser's cache and history, using spyware etc., so they become less concerned about their privacy infringement.

Based on our research findings, perceived ease of use does not impact behavioral intention to use the actual BBS unlike in our previous study on the blockchain-based prototype model, where ease of use was significant in the initial stage (*Shrestha & Vassileva, 2019a*). This is because users perceive BBS, a user-friendly web application, easier to learn and operate. Based on representative literature such as *Liu et al. (2010)*, UI design is the most significant item that affects perceived ease of use. Users, instead of being more concerned about learning to use the system, are concerned about the usefulness and overall performance of the BBS. Previous studies by *Venkatesh et al. (2003)*, *Chan & Lu (2004)*, *Pikkarainen et al. (2004)*, *Roca, García & de la Vega (2009)* confirm that usability (ease of use) remains non-significant to develop an intention to use the system.

According to the results of our study, we deduce perceived ease of use and quality of BBS as significant predictors of the usefulness construct. When users find BBS easier to use and believe they can be skillful in using it, they will consider the system as more useful to improve their performance and productivity. This is also confirmed by previous studies (*Gefen, Karahanna & Straub, 2003*; *Liu et al., 2010*). In our system, SCS allows customers to set their data sharing preferences and receive incentives for sharing their data as per the smart contracts, while DSS guarantees companies that the customer data they access have integrity and confirm provenance. So, users of each system, who feel more satisfied with these features, develop a higher understanding of its perceived usefulness. Eventually, with positive feelings about the usefulness of the BBS, users develop a stronger behavioral intention to accept the system. Since the quality of system has an insignificant direct effect on the intention to use the system for DSS, its effect through perceived usefulness is found out to be a significant positive effect in our study which is per the suggestions made by *DeLone & McLean (1992)*.

Moreover, the empirical results of our study also confirm a significant positive effect of the users' attitudes on their intention to use the BBS and suggest that the most important antecedent of attitudes towards using BBS is trust which is also supported by the previous studies (*Bhattacherjee, 2002*), which confirms that the trust predictor significantly influences the user's decision to adopt the online services. Therefore, familiarity with the significance of the underlying blockchain technology and the honesty of the companies to keep its promises of protecting privacy, securing information and incentivizing customers for sharing their data bring a higher level of trust and stimulate positive attitudes of

customers towards using the SCS. Similarly, trusting the blockchain technology for its integrity and dependability significantly improves the company's attitudes towards adopting the DSS.

According to (*Shin, 2010*), trust has a moderating effect on perceived security and perceived privacy when it comes to adopting social networking sites. Perceived security has a mediating effect on perceived privacy that correlates to trust (*Rios, Fernandez-Gago & Lopez, 2017*). The findings from our study suggest that perceived security has a direct effect on trust in the context of the prototype model and DSS. Outside of this, there is no significant relationship between security and other constructs. Perceived privacy has a direct effect on user trust and perceived security, which reinforces the findings by *Rios, Fernandez-Gago & Lopez (2017)* that claims perceived security and perceived privacy are related. Based on our findings, the direct effect of perceived privacy on users' attitudes towards BBS is only significant for SCS and is moderated by trust in all pre-test, SCS and DSS models.

Our findings suggest that the influence of perceived privacy and perceived security depends strongly on which blockchain-based system users interact with. When answering the initial pre-test survey, participants have no system to base their ideas on. So, security, privacy, trust and BBS become abstract concepts. As abstract concepts, participants believe privacy affects security, security and privacy affect trust, and trust affects their intention to use the system. However, they are not aware of any direct effect of privacy and security on their choice to use the system.

In our study, we see that after using the SCS, there is a significant effect of perceived privacy on the user's attitude towards BBS. Yet, the pre-test and DSS survey results show that participants feel perceived privacy does not positively affect their attitudes towards BBS. Perceived privacy's effect on user's attitudes towards BBS is only significant with a customer-specific BBS like SCS but not significant with a company-specific BBS like DSS. However, trust has either a partial or complete mediating role in all kinds of BBS which is consistent with prior research (*Shin, 2010*).

Based on the initial pre-test survey results, we deduce that participants feel security protection mechanisms are an important indicator to trust the system. However, after using the SCS, we learn that perceived security tends to be an insignificant predictor of trust. For the DSS, the effect of perceived security on trust is once again significant. It may be because after experiencing the real-life blockchain-based system, respondents using the SCS becomes aware of the underlying security infrastructure of blockchain and smart contracts, but once they learn that the business process models deployed via smart contracts are committed on a public blockchain, they may care more about privacy and think less about underlying security. As they are not concerned about security, they want to have control over their data instead, the relative significance of perceived privacy to trust SCS for these users is higher. On the other hand, respondents experiencing DSS to access customer data may not care much about privacy since they are already putting their information through transparent processes for customers and other enterprises. Instead, they may care more about secure transactions, mitigating anomalies and malicious behavior in their consortium network and cyber-resilient smart contracts. Therefore,

perceived security may significantly affect trust in an abstract context, but with a specific context, it may be significant for a model like SCS and may not be significant for a model like DSS.

Prior research on the effect of perceived security and perceived privacy on user trust are mixed. *Shin (2019)* found a significant moderating effect of security on trust, but participants had no real interaction with a system. Studies on non-blockchain online services had comparable results. *McCole, Ramsey & Williams (2010)* found that perceived privacy and perceived security moderates the effect of trust. *Eastlick, Lotz & Warrington (2006)* empirically showed that the relationship between privacy concerns and trust was the third strongest of all relationships studied. *Chellappa & Pavlou (2002)* argued that perceived security is a stronger predictor of trust. All four of these studies were abstract and did not have participants engage with a real system before answering their survey. These results support our initial pre-test results. Without interacting with any system, participants often consider privacy, security, and trust to be strongly related.

In previous studies, where participants engaged with online services such as online shopping, perceived security had a stronger effect. Both (*Belanger, Hiller & Smith, 2002*) and (*Kim, Steinfield & Lai, 2008*) found that perceived security had a stronger effect than perceived privacy on consumer behavior. *Roca, García & de la Vega (2009)* found that perceived privacy did not influence trust, but they did not consider the influence of security factors moderating privacy concerns in their model based on extended TAM. These do not align with our findings from when participants used the SCS. Our study found security has no significant relationship to trust in SCS, while privacy significantly affects trust and attitudes towards BBS.

The discrepancy between results from abstract studies and studies with concrete systems shows how important it is to focus on the latter. Although the abstract studies show there was a strong relationship between trust, privacy, and security, the studies with actual eCommerce systems have mixed and inconclusive results. Furthermore, studies on eCommerce systems focus on the customer. Few relevant studies focus on the company's trust and its intention to use the technology. Therefore, we cannot find other results to compare to the current study's finding that for DSS perceived security positively affects trust in BBS, and trust completely mediates the influence of privacy on attitudes towards adopting the BBS. Also, based on our pre-test and post-test results, there is no mediating effect of security over the perceived privacy on the users' attitudes towards BBS. Further study is needed with specific types of BBS to see if there are more BBS types other than customer-specific and company-specific and to better understand which trust construct is significant for each type of system.

Our study also brings a methodological contribution to the literature with the use of partial least square structural equation modelling (PLS-SEM) to analyze the user acceptance of the concrete blockchain-based application. PLS is component-based and can model the latent constructs under conditions for smaller sample sizes by maximizing the explained variance of dependent indicators and use multiple regressions to observe the effect of predictors on the response variables (*Chin, Peterson & Brown, 2008*; *Hair et al.,*

*2013*). Furthermore, this study contributes to the methodology by adopting Dillon-Goldstein's rho, for estimating internal consistency reliability, which is suggested as an always better choice than conservative Cronbach's alpha in the presence of skew items and smaller samples (*Demo et al., 2012*).

## LIMITATIONS

The main limitation of our study is that our findings are based on smaller targeted population size and only on two specific types of BBS. Therefore, the results may not generalize to the broader population and to any type of BBS. Further study may consider using a larger sample with specific types of BBS; to explore BBS types other than customer-specific and company-specific, and to better understand which trust antecedent is significant for each type of system. Moreover, an obvious limitation comes from using the same participants for both systems. With most respondents who participated in the DSS study also completed the SCS study, they also satisfy the inclusion criteria of DSS while doing the SCS study. The DSS study had participants only with a technology background. Also by taking on separate roles, participants may have experienced different motivations that skewed their survey results, so further study is needed to draw any conclusions about the role users take and what factors influence their desire to use the specific BBS. Likewise, the results showed that almost 79% of participants for SCS had a basic knowledge of blockchain technology while 19% had advanced knowledge and some of the participants belonged to academia. To address this, we need to consider an underlying effect of participants' background on their behavioral intention to use BBS. Therefore, this study offers an opportunity for future exploration of BBS to consider multigroup analysis based on participants' demography and background knowledge when analyzing the endogenous and exogenous variables, which will further explain the user acceptance of the BBS.

## CONCLUSIONS

In this paper, we presented the augmented TAM with trust model on our real-life blockchain-based system (BBS), which comprises two subsystems: Shopping Cart System (SCS) and Data Sharing System (DSS). The main contribution of our study to the body of knowledge is that, to the best of our knowledge, this study is the first to examine the augmented TAM with trust model using real-life concrete blockchain-based applications. The empirical study validated our research model and supported most of the research hypotheses that we set based on our research. Our findings suggested that TAM-based predictors and trust constructs cannot be applied uniformly to BBS. Depending on the specifics of the BBS, the relationships between perceived trust, perceived security, perceived privacy and attitudes towards the system might change. In SCS trust was the strongest determinant of attitudes towards the system, but in DSS, privacy was the strongest determinant of attitudes towards the system. Quality of system had the strongest total effect on intention to use SCS, while perceived usefulness had the strongest total effect on intention to use DSS. Trust significantly influenced the users' attitudes towards

both types of BBS, while security did not have any effect on users' attitudes toward BBS. In SCS, privacy positively affected trust, but security had no significant effect on trust, whereas, in DSS, both privacy and security significantly influenced trust. In both BBS, trust had a moderating effect on privacy that correlated directly with attitudes towards BBS, whereas security had no mediating effect between privacy and attitudes towards BBS. Hence, we recommend that while implementing and upgrading blockchain-based solutions, the decision-makers should carefully consider the trust patterns and address the associated privacy challenges of the users. Designers and decision-makers for the industries should know that the effect of trust antecedents is context-dependent whether it is customer or company-oriented. For the development of customer-oriented BBS, the effect of a privacy-aware system to influence users' attitudes toward BBS is relevant. For the development of a company-oriented BBS, additional security measures must also be carefully addressed to significantly influence users' trust in BBS, which in turn positively leads to a higher intention to adopt the system. In future work, we plan to investigate multigroup analysis based on participants' background knowledge when analyzing the latent variables and perform the qualitative analysis based upon the respondents' feedback, which will further explain the user acceptance of the BBS.

## ACKNOWLEDGEMENTS

We thank Mr. Kiemute Oyibo of the University of Waterloo for his helpful discussion and consultation about the correctness of the statistical analysis.

### Funding
This work has been supported by the University of Saskatchewan Dean's Scholarship and Teacher-Scholar Doctoral Fellowship to the first author and by the Natural Sciences and Engineering Research Council Discovery Grant of the second author. The funders had no role in study design, data collection and analysis, decision to publish, or preparation of the manuscript.

### Grant Disclosures
The following grant information was disclosed by the authors:
University of Saskatchewan Dean's Scholarship and Teacher-Scholar Doctoral Fellowship.
Natural Sciences and Engineering Research Council Discovery Grant.

### Competing Interests
Ajay Kumar Shrestha is a Ph.D. candidate at the University of Saskatchewan. Ajay Kumar Shrestha has received the UofS CGPS Dean's Scholarship, Teacher-Scholar Doctoral Fellowship, NSERC (Discovery grant of Dr. Vassileva). Ajay Kumar Shrestha is a member of the Services Society.

Julita Vassileva is an Academic Editor for PeerJ.

## Author Contributions

- Ajay K. Shrestha conceived and designed the experiments, performed the experiments, analyzed the data, performed the computation work, prepared figures and/or tables, authored or reviewed drafts of the paper, and approved the final draft.
- Julita Vassileva conceived and designed the experiments, authored or reviewed drafts of the paper, and approved the final draft.
- Sandhya Joshi performed the computation work, authored or reviewed drafts of the paper, and approved the final draft.
- Jennifer Just analyzed the data, authored or reviewed drafts of the paper, and approved the final draft.

## Ethics

The following information was supplied relating to ethical approvals (i.e., approving body and any reference numbers):

The University of Saskatchewan Behavioural Research Ethics Board (Beh-REB) granted Ethical approval to carry out the study (Ethical Application Ref: Beh # ID2106).

## Data Availability

The anonymized raw datasets, and the aggregated data, are available in the Supplemental Files.

## Supplemental Information

Supplemental information for this article can be found online at http://dx.doi.org/10.7717/peerj-cs.502#supplemental-information.

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
