# Peer review of "Augmenting the technology acceptance model with trust model for the initial adoption of a blockchain-based system"

_PeerJ Computer Science, doi:10.7717/peerj-cs.502_

## Round 0.1 · original submission · Major Revisions

The reviewers have suggested a mix of minor and major reviews. In addition to, but also coinciding with some of the issues the reviewers have indicated, I will also suggest that the paper provide substantial details on how this work adds to their previous work (Shrestha & Vassileva, 2019a). Particularly, highlight if there is anything new in the methodology and findings, or whether it is a similar study with a new questionnaire and with new respondents. So to say, what are the new ideas in this paper that are 'useful/reusable in general' for the broader community, that did not already exist in the previous paper?

Two other crucial limitations of the current work are that (i) it is based on a specific system the authors themselves had designed, and (ii) the lack of diversity among the respondents. For the inferences to be meaningful, it would be desirable to expand the scope of the work and address these concerns (at least to some extent, if not in its full generality).

Reviewer 1 ·

Basic reporting

In the manuscript, the authors report findings of an extended technology acceptance model (TAM) applied to analysis of blockchain use cases.

The authors provide detailed background and context in the paper covering both TAM, blockchain, and research methodology. Motivation of the work is clear.

The paper includes well formulated hypotheses to be tested through survey studies of targeted users.

The extended TAM for blockchain based systems focuses on constructs, security, privacy, and trust. Because these concepts are closely intertwined, I feel that additional elaboration of their interpretation in context of blockchain based systems is useful to understand the structural model.

Experimental design

The survey study is designed to validate specific hypotheses derived from the new TAM for blockchain based systems. There are two targeted use cases (one consumer oriented – shopping, and the other one enterprise oriented – data sharing). The results show that TAM based predictors and constructs are affected by the use case differences.

The authors conduct a thorough analysis of the data using established structural model scheme. I feel that some of the findings are informative. Although studies using similar methodology have been performed in the past for other ICT applications, there is a gap of similar analysis to blockchain based systems, which is tackled by the authors’ work. The analysis seems comprehensive.

I wonder if the path weighting structural modeling approach has been well validated. How to separate correlational vs causal relationships between the TAM constructs? Some clarification may be helpful.

Validity of the findings

The authors conduct a comprehensive analysis of the data. The results are provided at the end of the paper.

There are certain limitations on how the results could be generalized, which is also discussed by the authors.

Additional comments

Overall, the work focuses on validating a decision model for adopting blockchain based systems. The model incorporates constructs important to blockchain based systems such as security, privacy, and trust as predictors. It applies well established methodology in technology adoption studies. Some of the reported findings are informative and potentially useful for designing new blockchain based systems.

A limitation is that the surveyed users are overall well educated and have better knowledge in ICT than the general public. Likely literacy levels for both computer and blockchain play important roles in the decision making process of adoption. It is not clear how the TAM could model such effect.

Additional justification of applying path structural modeling scheme to validate decision model of technology adoption could be useful to readers who are not experts in this area.

Reviewer 2 ·

Basic reporting

The paper reports an empirical study investigating how various latent variables impact users’ attitudes and intentions of using a blockchain-based system. The study has used the primary Technology Acceptance Model (TAM)-based predictors. Data from more than 60 respondents, mostly Master level IT students, are collected and analyzed.
Positive aspects of the study and the manuscript
- Comprehensive related studies
- Detailed comparison of this study’s results with related work
- Structured presentation of the research design
- A detailed report of the data analysis results

Experimental design

The experimental design is explained in detail. However, I could not find a mapping between the constructs and the questions in each questionnaire. From the questions I can see in the questionnaires, constructs of two latent variables need to be explained better.
- Quality of the system. In the software engineering domain, security and privacy are regarded as part of Quality of the system. It seems that the questions in the questionnaire related to this latent variable are about dependability. In this case, it would be better to use reliability/dependability rather than using the Quality of the system, which is a broad concept.
- Attitude: I could not find myself which constructs measure attitude in the questionnaire. The term attitude also needs to be defined more precisely in the paper.

Validity of the findings

In the “participants and demographics” chapter, the authors mentioned that some data are cleaned. Please explain the criteria applied to include/exclude data.

The main limitation of the study is the limited number of respondents. The respondents are also people who know IT technologies well. The study needs to be replicated to users of
different profiles. The authors have discussed these limitations. This is acceptable.

Another limitation the authors need to discuss is the results related to DSS. DSS is more related to companies than individual users. How much can we trust the results from students, who usually do not have industrial experience, to answer company-related questions? This needs to the reflected and discussed in the paper.

Additional comments

The discussion chapters can be structured better. Most content of the discussion chapter is about comparison with related work. The authors could discuss more the implications of the results to academia and industry.
In the discussion chapter, some conclusions are too strong. For example, line 870, it is because … Without follow-up interviews to confirm the assumptions, the authors shall write “it may be because …”

·

Basic reporting

1) They constructed the survey questions after a pilot study with 14 researchers, so the questionnaire is supposed to be good enough for the study.

2) The authors evaluated ease of use, perceived usefulness, quality of the system, intention to use the system, security, trust, and other factors that seem promising.

Experimental design

It is review/study type manuscript rather than technical paper. The experimental design are not focused which looks fine considering its nature.

Validity of the findings

1) 14 hypotheses are constructed and discussed their decision with the required illustrations.

Additional comments

1) Authors have merged the related works within the introduction, which looks verbose but lacks focus on the problems, challenges targeted and specific contributions claimed.

2) It is recommended to separate/merge the referred literature within the later 'Related Work subsection and keep the introduction focused on the targeted problems and challenges. In addition, it is better to have a list of contributions Authors claim after addressing the issues.

3) The contents style as in line 125, 127, 146, etc., preceded with '..our/current study..' looks bringing an impression that Authors have struggled to fit the survey type literature in the Blockchain-based technical contexts (Model for the initial adoption of a blockchain-based
system) as mentioned in the Abstract.

4) Writing needs improvement in terms of professionalism and broader readers' attention.

5) Too many mathematical notations and values seem confusing average readers and conflict and moving its focus from review-nature.

6) Language need to improve, should check grammatical errors and typos, formatting.

7) Its highly recommended to improve the organization of the manuscript.

---

## Round 0.2 · accepted · Accept

The reviewer with major revision recommendation in the first round has deemed the revision acceptable, and as such, I recommend accepting the paper.

Reviewer 2 ·

Basic reporting

The updated manuscript has addressed the comments from the reviewers. I do not have further comments. I think the paper is acceptable.

Experimental design

The experimental design is reasonable.

Validity of the findings

The findings are valid and the limitations are discussed properly.

Additional comments

The paper can be accepted from my point of view.